# Impact of climate change on maternal health outcomes: An evidence gap map review

**Salima Meherali**[1]*, **Saba Nisa**[1], **Yared Asmare Aynalem**[1], **Megan Kennedy**[2], **Bukola Salami**[3], **Samuel Adjorlolo**[4], **Parveen Ali**[5], **Kênia Lara Silva**[6], **Lydia Aziato**[4], **Solina Richter**[7], **Zohra S. Lassi**[8]

1 College of Health Sciences, Faculty of Nursing, University of Alberta, Edmonton, Canada, 2 John W. Scott Health Sciences Librarian, Walter C. Mackenzie Health Sciences Centre, University of Alberta Library, Edmonton, Canada, 3 Department of Community Health Sciences, Cumming School of Medicine, University of Calgary, Calgary, Alberta, Canada, 4 Department of Mental Health, School of Nursing and Midwifery, College of Health Sciences, University of Ghana, Legon, Accra, Ghana, 5 School of Allied Health Professions, Nursing and Midwifery, University of Sheffield and Doncaster and Bassetlaw Teaching Hospital Trust, Sheffield, United Kingdom, 6 Department de Enfermagem Aplicada, Escola de Enfermagem, Universidade Federal de Minas Gerais, Belo Horizonte, Minas Gerais, Brasil, 7 College of Nursing, University of Saskatchewan, Saskatoon, Canada, 8 School of Public Health, Faculty of Health and Medical Sciences, The University of Adelaide, Adelaide, Australia

* meherali@ualberta.ca

**Data Availability Statement:** Data is freely available in the manuscript itself and uploaded supplementary files.

## Abstract

Climate change poses unique challenges to maternal well-being and increases complications during pregnancy and childbirth globally. This evidence gap map (EGM) aims to identify gaps in existing knowledge and areas where further research related to climate change and its impact on maternal health is required. The following databases were searched individually from inception to present: Medline, EMBASE, and Global Health via OVID; Cumulative Index to Nursing and Allied Health Literature (CINAHL) via EBSCOhost; Scopus; and organizational websites. In this EGM, we integrated 133 studies published in English, including qualitative, quantitative, reviews and grey literature that examined the impact of climate change on maternal health (women aged 15–45). We used Covidence to screen studies and Evidence for Policy and Practice Information (Eppi reviewer)/Eppi Mapper software to generate the EGM. Data extraction and qualitative appraisal of the studies was done using critical appraisal tools. The study protocol was registered in International Platform of Registered Systematic Review and Meta-analysis Protocols (INPLASY) # INPLASY202370085. Out of 133 included studies, forty seven studies were of high quality, seventy nine moderate equality and seven low quality. This EGM found notable gaps in the literature regarding the distribution of research across regions. We found significant research in North America (51) and Asia (40 studies). However, Africa and the Caribbean had fewer studies, highlighting potential disparities in research attention and resources. Moreover, while the impact of extreme heat emerged as a prominent factor impacting maternal well-being, there is a need for further investigation into other climate-related factors such as drought. Additionally, while preterm stillbirth and maternal mortality have gained attention, there is an overlook of malnutrition and food insecurity indicators that require attention in future research. The EGM identifies existing research gaps in climate change and maternal health. It emphasizes the need

**Funding:** Author (SM) received funding from World Universities Network Research Development Funds (WUN RDF) for this project. Grant # RES0061104. The funders had no role in study design, data collection and analysis, decision to publish, or preparation of the manuscript.

**Competing interests:** The authors have declared that no competing interest exist.

for global collaboration and targeted interventions to address disparities and inform climate-responsive policies.

## Introduction

The World Health Organization (WHO) refers to maternal health as women's health during pregnancy, childbirth, and postpartum [1]. According to a report by the WHO [2], an estimated 287,000 women die due to maternal health complications each year globally, with almost 800 maternal deaths occurring daily, accounting for death every two minutes. Furthermore, regional disparities in maternal health are evident across the world. In 2020, Europe and Northern America, Eastern and South-Eastern Asia, Northern Africa, Western Asia, Latin America, and the Caribbean exhibited low maternal mortality rates below hundred. However, Sub-Saharan Africa alone contributed to about seventy percent of global maternal deaths in 2020, with Central and Southern Asia accounting for nearly seventeen percent [2]. Climate change increases the vulnerability to adverse maternal health outcomes. Mothers, especially those in resource-limited settings, face the heightened impact of climate change during the stages of pregnancy, childbirth, and the postpartum period [3]. Climate change negatively impacts maternal health during pregnancy, causing thermo-regulatory challenges due to internal heat production from fetal growth [4]. Exposure to the effects of climate change at the early stage of fetal development can cause immediate harm or damage that becomes evident later in life, resulting in lasting effects over a lifetime and even over generations [5].

Climate change affects maternal health in many ways, impacting various aspects of well-being through both direct and indirect mechanisms. Physiological risks such as extreme heat leads to dehydration by excessive sweating in pregnant women and causes the onset of early labour [6]. High temperatures can also increase the risk of preterm birth, low birth weight, abortion, and stillbirths.[3] Extreme cold and earthquake exposure in pregnancy has been associated with maternal hypertension and preeclampsia [7–9], and gestational diabetes [10]. Climate change indirectly affects mental health by causing anxiety and depression due to uncertainty about the environment, worries about how it will impact their children in the long term [11, 12]. During pregnancy, the energy demand of women increases by approximately twenty percent, which also continues throughout the period of breastfeeding [13]. However, the socioeconomic challenges posed by climate change exacerbate issues related to food insecurity and limit access to antenatal and postnatal care [14]. The impact of climate change, mainly through events like drought, extends beyond food security, leading to significant consequences such as livestock deaths, crop failure, and severe malnutrition [13]. Furthermore, climate impacts the mental health of mothers and their children in their growth and development [5].

Over the past five years, the world has witnessed increased climate change events such as extreme heat, wildfires, and droughts [15–18]. The impacts of climate change exhibit geographical variations, affecting regions, countries, and specific locations differently. Vulnerable regions face higher risks, as research indicates that 3.6 billion people reside in areas highly susceptible to climate change [2]. The vulnerability of regions has escalated over time, with increases observed from 1990 to 2017 in the African region (28.4% to 31.3%), Southeast Asia region (28.3% to 31.3%), and the Western Pacific region (33.2% to 36.6%) (19). In 2020, 770 million people faced hunger, primarily in Africa and Asia, and vector-borne diseases posed a significant threat, causing over 700,000 annual deaths [2]. Without preventive actions, the

WHO conservatively estimates an additional 295,000 yearly maternal deaths by 2030 due to climate change impacts on diseases like malaria and coastal flooding [2]. Furthermore, differentiation in geographical location, such as high temperature, increases the risk of heat shock and will be most prominent in already hot countries and for people with physically demanding labour [13].

While the effects of climate change on maternal physical health, psychological health, and food security have garnered attention in specific regions, there is a lack of understanding of how climate change affects maternal well-being in diverse geographical contexts. This review aims to bridge this gap by systematically mapping the global evidence available on the impact of climate change on maternal well-being by generating the evidence gap map (EGM).

## Why it is crucial to develop an EGM

An EGM is a systematic and visual representation of the existing evidence, or lack thereof, on a specific research question or topic. EGMs help researchers, policymakers, and practitioners make informed decisions about where additional research efforts are required to address gaps in understanding or implementation [19]. EGMs typically represent and categorize studies based on specific criteria such as the number of studies, study designs, intervention/ exposure, outcomes, and quality appraisal of the included studies [19]. Additionally, the EGM aids in avoiding duplication of research efforts and resources, fostering collaboration, and maximizing the impact of interventions [19]. We conducted an EGM on climate change's impact on maternal health following Campbell Standards (S1 File). Our EGM provides access to the available research evidence on the impact of climate on maternal health outcomes. Each row corresponds to different exposures or events associated with climate change, while the columns outline the outcomes related to the impact of climate change on maternal health. The segmenting section provides information on the number of included studies, the qualitative appraisal of these studies, and the study designs.

The study protocol was registered with International Platform of Registered Systematic Review and Meta-analysis Protocols (INPLASY) # INPLASY202370085.

## Methods

### Search strategy

The search strategy for this EGM is reported in adherence to the Preferred Reporting Items for Systematic Reviews and Meta-Analyses for Protocols: Extension for Scoping Reviews for Searching (PRISMA-S) extension 22 (Fig 1) and followed Prisma reporting checklist (S2 File). The search strategy was developed by an experienced health sciences librarian at the University of Alberta (M.K.) in consultation with the research team. The following databases were searched individually from inception to (13 January 2024): Medline, EMBASE, and Global Health via OVID; Cumulative Index to Nursing and Allied Health Literature (CINAHL) via EBSCOhost; Scopus; and organizational websites. The search strategy focused on maternal health-related concepts, including maternal health, maternal mortality, and perinatal death. It also explored concepts related to maternal health services and pregnancy care. The search incorporated terms associated with pregnancy, labour, obstetrics, pregnancy outcomes, complications, and various maternal disorders. Additionally, it addressed climate change-related concepts such as the carbon cycle, global warming, droughts, floods, and extreme weather events. The strategy excluded non-human studies (S3 File). A total of 6895 studies were initially identified from various databases. Using Covidence software and manual review, duplicates were carefully removed. Subsequently, 300 studies underwent screening based on eligibility c criteria, with 133 studies ultimately included in this review.

PRISMA 2020 flow diagram for new systematic reviews which included searches of databases, registers and other sources

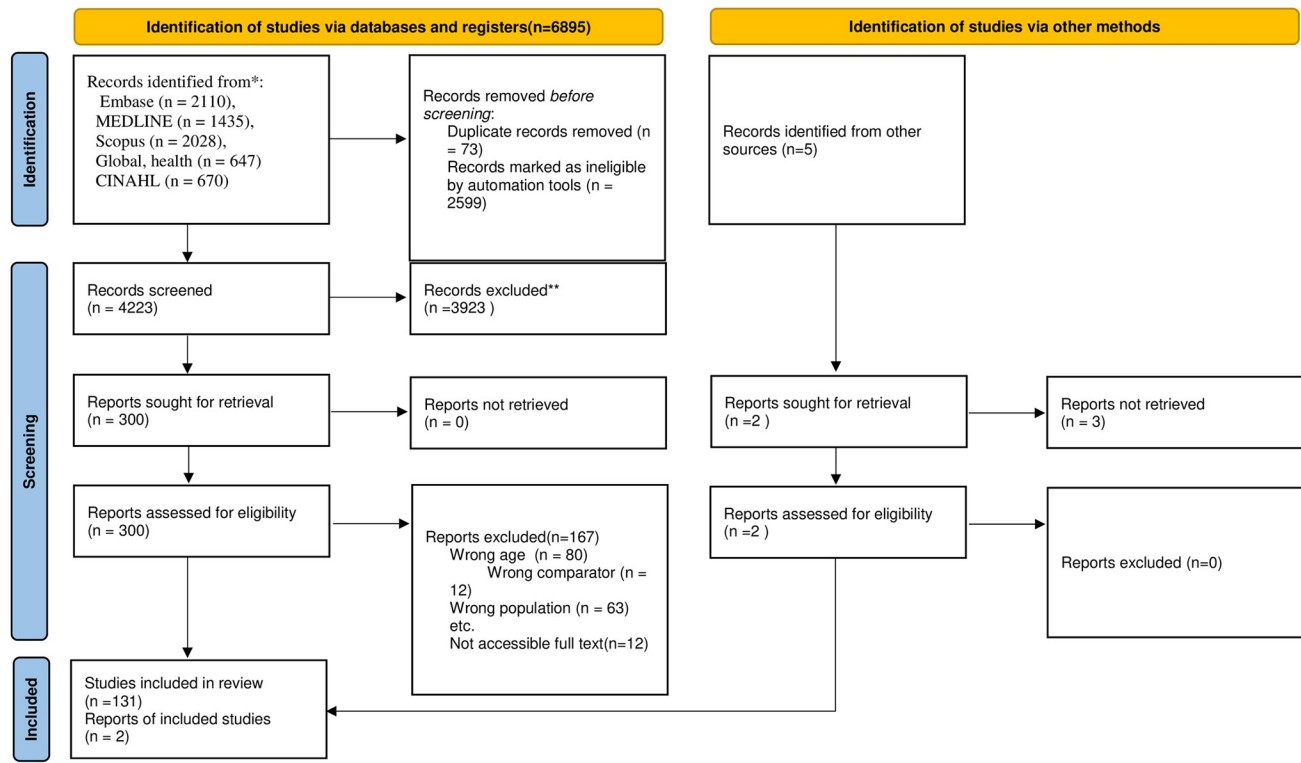

**Fig 1. Prisma flow chart for climate change impact on maternal health.**

## Selection criteria

Our selection criteria focused on primary studies (qualitative and quantitative), reviews (scoping, systematic), and grey literature that specifically examined the impact of any type of climate change on maternal health. We included studies published in English, with the target population being women aged 15–49. The review considered studies conducted globally, regardless of settings or context. Due to language limitations, only literature published in English was deemed eligible for inclusion. We excluded studies that did not focus on maternal health aged 15–49 and those published in languages other than English. Inaccessible full-text articles were also excluded after contact with the authors.

## Screening and data extraction

All studies identified from the databases were imported into Covidence, an online screening software [20]. We used Covidence for title abstract screening and full-text screening. Two reviewers, SN and YA, independently completed the screening phase, with any conflicts resolved by a third reviewer, SM. Articles meeting the final inclusion criteria were exported to EndNote as RIS files [21] and then transferred to Evidence for Policy and Practice Information (EPPI) Reviewer for coding [19]. Finally, a total of 133 studies were included in our study. Data extraction was done using an EPPI reviewer software (version 6.15.0.2). Data extraction

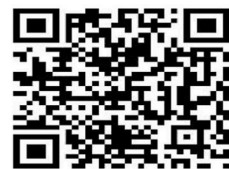

Link to map: Climate change and maternal health-EGM   OR Scan

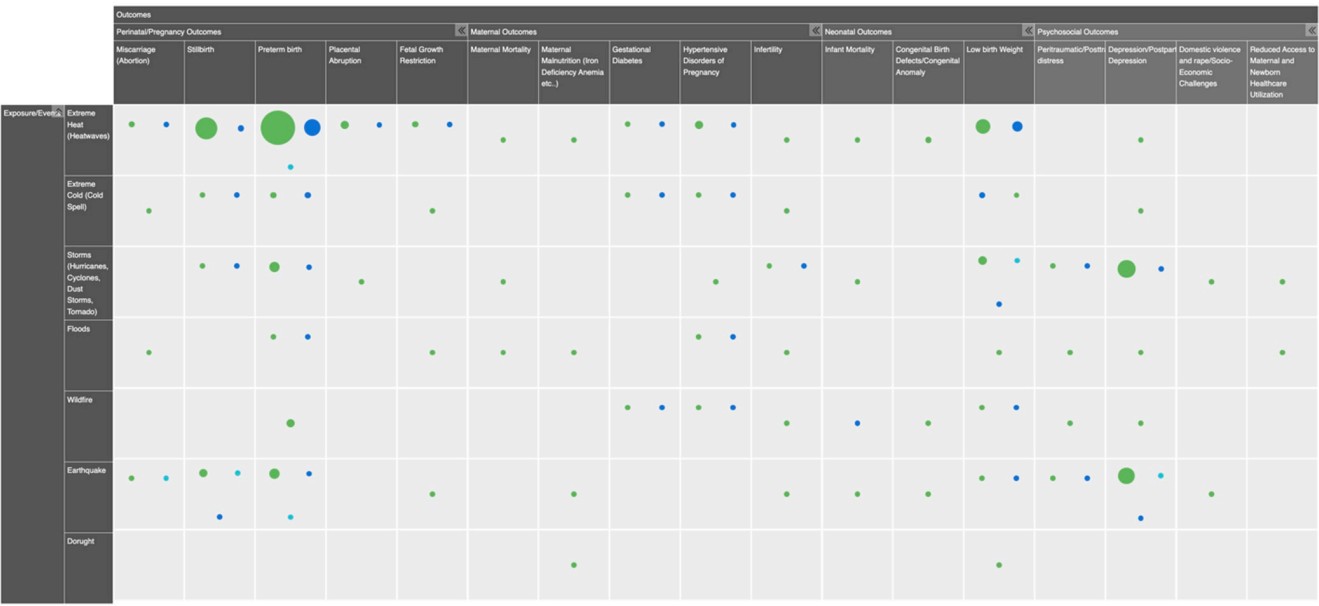

**Fig 2. Evidence gap map: Climate change impact on mothers.**

as a table was also done in Word to enhance transparency (S4 File). We used Eppi Reviewer software to create an EGM (Fig 2). The software helped us code detailed information, including intervention outcomes and quality appraisals for all the studies. This EGM can be accessed by scanning QR code mentioned in Fig 2 and this data can be used to identify areas of available evidence related to climate change and maternal health outcomes globally.

## Quality appraisal

Critical appraisal of all the included studies was done using specific tools: Assessment of Multiple Systematic Reviews 2 (AMSTAR) for systematic reviews (RCTs and non-RCTs), the Mixed Methods Appraisal Tool (MMAT) for primary studies (qualitative and quantitative), and a Qualitative Meta-Review Quality Assessment Tool for the qualitative synthesis. Out of the total one hundred and thirty-three studies, forty-seven were classified as high quality, indicating a robust methodology and reliability of results. Additionally, seventy-nine studies were deemed moderate quality, suggesting satisfactory methodological rigor. On the other hand, seven studies were categorized as low quality, implying potential limitations in study design or

execution. This quality assessment provides a comprehensive overview of the reliability and strength of the evidence synthesized in our research.

To aid readers in understanding specialized terminology, a glossary is provided at the end of this paper, offering concise definitions of key terms for climate change events and outcomes used in our study.

### Findings

The findings of this EGM highlight critical gaps in evidence. The details of the gaps are discussed below (Fig 3).

### Evidence mapping of publication trends across years

Our findings reveal a positive trend in the research landscape, demonstrating a growing focus on the impact of climate change on maternal health from 2012 to 2023. Starting with two publications in 2012, the numbers have steadily increased, reaching a peak of twenty-seven publications in 2022. The significant surge to twenty-seven publications in 2022 suggests heightened research interest and dedication to understanding the complex dynamics. While

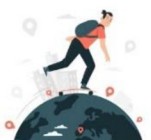
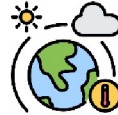

Regional Disparities

Lack of research evidence in Africa and Caribbean

Gaps in Events/Exposure

Limited research on droughts, Landslide, snow fall

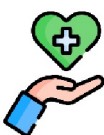
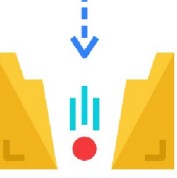
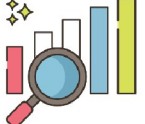

Evidence Gaps

Gaps in Outcomes

Scarcity of research on climate change impact on breast feeding, access to contraception, exacerbation of preexisting health conditions and social support

Gaps in study designs

The need for increased attention to interventional and qualitative studies

**Fig 3. Evidence gaps in the area of climate change impact on maternal health.**

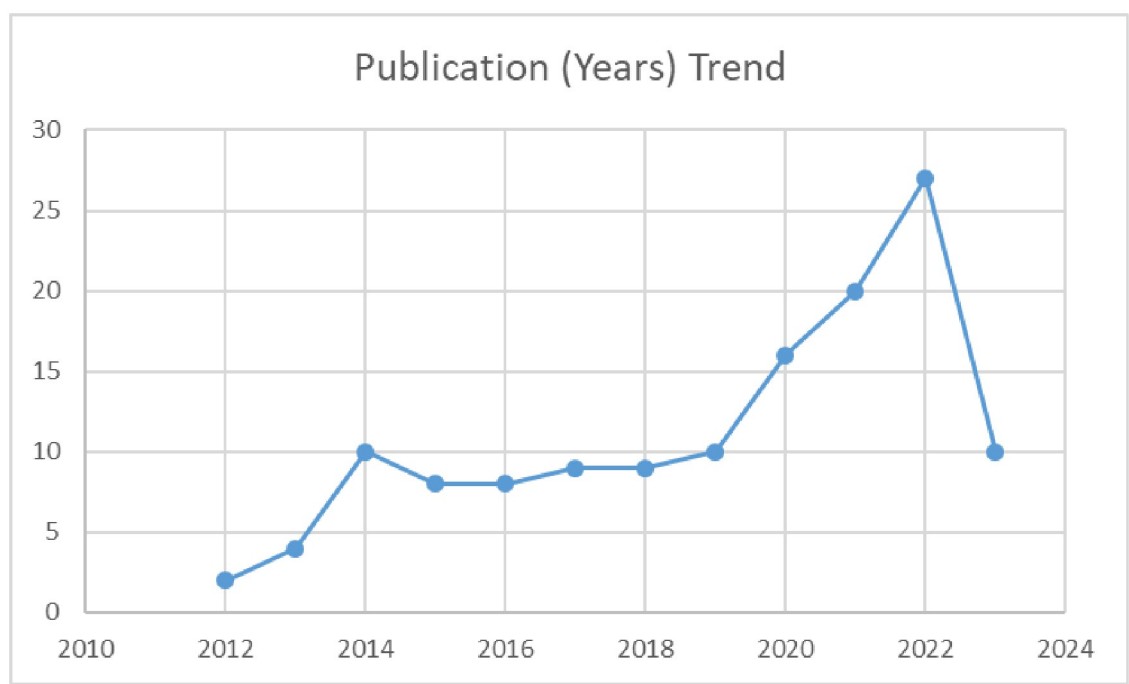

**Fig 4. Publication trends of included studies (years).**

there was a slight regression in 2023 with ten publications, the overall positive trend highlights a promising trajectory in exploring climate change's implications for maternal health (Fig 4).

### Disparities in evidence distribution across regions

We found a variation in the distribution of available evidence across regions. The EGM review revealed that Asia significantly contributes to the literature on the impact of climate change on mothers with forty studies, closely followed by North America with fifty-one. Europe adds to the discourse with nine studies, Oceania one, South America seven, the Caribbean six, and Africa with a comparatively lower count of five evidence. The observed concentration of research in North America and Asia suggests a heightened research emphasis in these regions on understanding the intricate interplay between climate change and maternal health. Conversely, Africa and the Caribbean emerge with fewer published studies, indicating a potential gap in research exploration in these areas. This geographical distribution of studies reflects the research priorities of different regions and raises important considerations regarding the global inclusivity of evidence. For a visual representation of this distribution, refer to (Fig 5).

### Findings by climate change events

This EGM (Fig 2) delves deeply into the repercussions of climate change on maternal health, covering a spectrum of occurrences like extreme heat, cold, storms, floods, wildfires, earthquakes, and droughts. The comprehensive analysis reveals a significant research study conducted on extreme heat stands out with eighty-seven studies, twenty-four studies on extreme cold, earthquakes follow closely with thirty-nine studies, while storms, including hurricanes, cyclones, dust storms, and tornadoes, draw attention with thirty-two studies, providing insights into their effects on maternal health dynamics. The narrative shifts to wildfire (nineteen

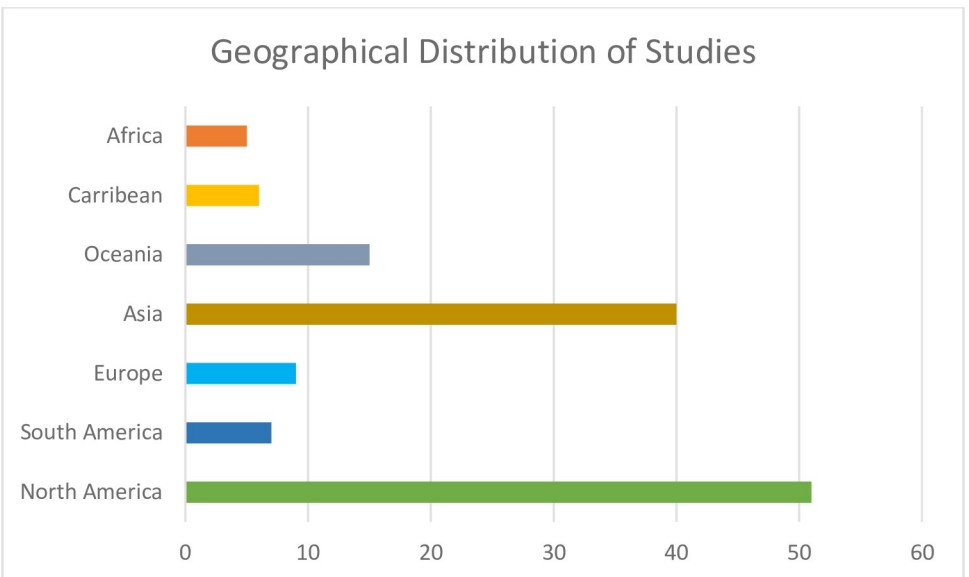

**Fig 5. Geographical distribution of studies.**

studies) and floods (sixteen studies), contributing to understanding water-induced impacts on maternal health. However, drought emerges as the least researched event, with only two studies dedicated to unravelling its subtle yet profound consequences on maternal health. These findings emphasize the urgency for inclusive research across all climate events, like droughts, to shape effective strategies for preserving maternal health amid a changing climate.

## Findings by outcomes

Regarding maternal health-related outcomes due to climate change, the studies predominantly focus on several crucial indicators. The EGM found a concentration of studies (eight) investigating the impact of climate change on the incidence of miscarriage, shedding light on the susceptibility of pregnancies to environmental shifts. Additionally, stillbirth is extensively covered in twenty-six studies, while preterm birth is a focal point in sixty studies. Other outcomes under scrutiny include placental abruption (six studies), fetal growth restriction (eight studies), maternal mortality (three studies), malnutrition (four studies), gestational diabetes (eight studies), hypertensive disorders of pregnancy (thirteen studies), infertility (seven studies), neonatal mortality (four studies), congenital birth defects (five studies), low birth weight (thirty-two studies), peritraumatic/posttraumatic stress (six studies), depression/postpartum depression (twenty seven studies), domestic violence and rape (three studies), and reduced access to service utilization (three studies). Despite the existing evidence illuminating the impact of climate change on maternal health, there is a critical need to explore specific areas that have received insufficient attention. For instance, there is a notable dearth of research focusing on mothers with pre-existing chronic conditions. There is a notable absence of intervention studies aimed at improving the impact of climate change on maternal health. Moreover, research on climate change impact on breast feeding patterns, access to contraception and impact on social support is lacking. The lack of comprehensive research into effective interventions hampers our ability to develop strategies to enhance pregnant individuals' well-being in the face of environmental challenges. The shortage of such intervention studies highlights a critical gap in knowledge, emphasizing the urgent need for targeted research efforts.

## Discussion

The findings of the EGM identify and categorize existing evidence, highlighting both areas of knowledge and research gaps. The observed upward trajectory in the number of studies examining the impact of climate change on maternal health, with a peak in 2023, as illustrated in Fig 4, aligns with the growing global awareness of the intricate relationship between environmental shifts and maternal health [19, 22, 23]. This surge in research activity suggests an increasing recognition of the need to explore and comprehend the multifaceted implications of climate change on maternal health. The geographic distribution of these studies in our EGM unveils disparities across different regions (Fig 5). Notably, North America and Asia emerge as focal points, boasting fifty-one and forty studies, respectively. This concentration may be attributed to these regions' diverse climate events, population sizes, and research capacities [24, 25] underscoring the significance of understanding region-specific nuances in the climate-maternal health discourse. Conversely, the limited representation in Africa and the Caribbean, with only five and 6 studies, respectively, draws attention to potential research gaps. The scarcity of studies in these regions may indicate challenges such as limited research infrastructure funding disparities or a potential underestimation of the impact of climate change on maternal health in these specific contexts [3, 26–28]. While the increasing number of studies reflects progress, the regional disparities emphasize the imperative of a more equitable distribution of research efforts. A nuanced understanding of the impact of climate change on maternal health necessitates collaborative and region-specific investigations, ensuring that interventions and policies are effectively tailored to address the distinct challenges mothers face in different parts of the world.

The number of studies on extreme heat, earthquakes, and storms reflects a commendable effort to understand the complexities of these events and their implications on maternal health. The substantial attention given to extreme heat aligns with recognizing rising temperatures as a critical factor in climate change, with numerous studies emphasizing the adverse effects on maternal health [29–31]. Similarly, the focus on earthquakes and storms, including hurricanes, cyclones, dust storms, and tornadoes, acknowledges the urgency of addressing the maternal health risks associated with natural disasters. Existing literature has consistently highlighted the vulnerability of pregnant women during such events, emphasizing the need for targeted interventions and disaster preparedness strategies [32–35]. However, the EGM reveals a significant void in the research landscape regarding the consequences of drought on maternal health. With only two studies addressing this aspect, there is a glaring lack of understanding regarding the subtle yet profound impacts of prolonged water and food scarcity on pregnant women. This gap is concerning, given the increasing frequency and severity of drought events associated with climate change [36–38].

In the discourse surrounding maternal health outcomes influenced by climate change, a notable imbalance emerges in the attention afforded to specific dimensions. While mental health outcomes, stillbirth, and preterm birth have garnered significant focus, there exists a discernible lack of attention to crucial facets such as maternal mortality, malnutrition, food insecurity, and the heightened risks of domestic violence and rape. Maternal health directly influences child health. Ignoring these concerns may result in adverse effects on the well-being and development of children [39]. Malnutrition resulting from food insecurity increases the risk of complications during pregnancy, such as gestational diabetes, anemia, and low birth weight [40]. Domestic violence and sexual assault can have significant implications for reproductive health. Unplanned pregnancies, sexually transmitted infections (STIs) [41], and long-term reproductive health issues may arise, requiring appropriate medical attention. Future research should prioritize investigating the specific challenges posed by drought, exploring the

intricate links between water and food scarcity, maternal well-being, and pregnancy outcomes. Moreover, the identified gaps in our study regarding certain outcomes, such as malnutrition, domestic violence, and reduced access to service utilization, underscore the need for more comprehensive investigations into these less-explored facets.

## Implications

Efforts should prioritize inclusive studies, addressing vulnerabilities of marginalized populations, for a more nuanced understanding and tailored interventions in the face of climate change affecting maternal health globally. Furthermore, integrating climate change's impact on maternal health into educational programs ensures that future healthcare professionals are well-prepared to address emerging challenges. It also supports the idea of collaboration with urban planners to design heat-resilient urban environments with consideration given to green spaces, shade provision, and climate-sensitive infrastructure to mitigate the impact of high temperatures on pregnant individuals. The intersection of climate change and maternal health demands proactive adaptation strategies, resilient healthcare systems, and global efforts to mitigate the environmental factors contributing to these adverse outcomes. Moreover, programs related to mental health of mothers and their children should be designed to meet their needs in the era of climate change.

This EGM acknowledges several limitations that merit consideration. The decision to conduct searches exclusively in English, introduces a potential language bias. The exclusion of searches in other languages may have led to the oversight of valuable evidence, potentially explaining the limited representation of studies related to the Caribbean and Central Africa. Additionally, some primary studies and systematic reviews lacked clarity regarding certain methodological aspects, hindering the confidence rating determined through the quality appraisal process. Enhanced methodological transparency in these instances could have improved the overall robustness of the EGM. These limitations underscore the importance of interpreting the EGM findings with a degree of caution and highlight areas for improvement in future research endeavours.

## Conclusion

While existing research has effectively addressed the maternal health impacts of specific climate events, the EGM underscores the urgent need for a more balanced exploration, particularly regarding the understudied consequences of climate change events such as drought, wildfire and floods in different regions. It is a call to action for researchers, policymakers, and public health practitioners to prioritize and support studies addressing the impact of climate change on maternal health. While this EGM effectively addresses the question of 'what exists,' there is a pressing need for additional interventional studies to delve deeper into determining the most effective strategies for enhancing maternal health in resilience to climate change. A more comprehensive examination of interventions and outcomes on a global scale is essential to identify best practices and inform robust strategies that can effectively address the intersection of maternal health in relation to climate change.

### Glossary of terms

#### Climate events

**Extreme Heat:** Involves heat waves, high temperatures, and ambient temperatures.

**Extreme Cold:** Encompasses occurrences such as low temperature (Cold), cold spells and ice storms.

**Storms:** Includes atmospheric disturbances, including hurricanes, typhoons, cyclones, and volcanic eruptions.

**Drought:** Unusually low precipitation that causes water scarcity.

**Floods:** Overflow of water onto land, typically from heavy rain or melting snow.

**Wildfire** Uncontrolled and rapidly spreading fire in vegetation, forests, or grasslands.

**Earthquake:** A sudden and pronounced shaking of the Earth's surface caused by the movement of tectonic plates beneath

**Outcomes**

**Maternal mortality:** The death of a woman during pregnancy, childbirth, or within 42 days after delivery

**Gestational Diabetes:** Diabetes that develops during pregnancy, typically in the second or third trimester, and is characterized by elevated blood sugar levels

**Hypertensive disorders of pregnancy:** A group of conditions characterized by high blood pressure during pregnancy, including gestational hypertension, preeclampsia, and eclampsia, which can pose risks to both the mother and the baby.

**Infertility:** Inability to carry a pregnancy

**Malnutrition**: A health condition resulting from an inadequate or imbalanced intake of essential nutrients, leading to physical and developmental issues.

**Peritraumatic/Posttraumatic Stress:** Emotional distress experienced during or after a traumatic event, potentially affecting mental well-being.

**Depression/Postpartum Depression:** Mood disorders characterized by persistent feelings of sadness or loss of interest, occurring either during or after pregnancy.

**Domestic Violence and Rape:** Physical, emotional, or sexual abuse within intimate relationships, including instances of non-consensual sexual assault.

**Reduced Access to Service Utilization:** Barriers or limitations preventing individuals from accessing necessary health or support services during pregnancy and childbirth.

**Miscarriage:** Unintended termination of a pregnancy before the 20th week, typically involving spontaneous expulsion of the fetus from the uterus.

**Stillbirth:** A baby is born without signs of life after 20 weeks of gestation.

**Preterm Birth:** The delivery of a baby before completing 37 weeks of pregnancy.

**Placental Abruption:** The premature separation of the placenta from the uterus before the baby is born, which can lead to bleeding and complications for both the mother and the baby.

**Fetal growth restriction:** A condition in which a fetus does not reach its expected size or fails to grow at the normal rate during pregnancy, potentially leading to health complications for the baby.

**Neonatal Mortality:** The death of a newborn within the first 28 days of life.

**Congenital Birth Defects:** Structural or functional abnormalities present at birth, often resulting from genetic or environmental factors affecting fetal development.

**Low Birth Weight:** The condition of a newborn weighing less than 2,500 grams (5.5 pounds) at birth.

## Supporting information

**S1 File. Campbell collaboration checklist for evidence and gap maps: Reporting standards.**
(DOCX)

**S2 File. PRISMA checklist.**
(DOCX)

**S3 File. Search strategy: Climate change and maternal health outcomes.**
(DOCX)

**S4 File. Data extraction for climate change and maternal health outcomes (EGM).**
(DOCX)

## Author Contributions

**Conceptualization:** Bukola Salami, Samuel Adjorlolo, Parveen Ali, Kênia Lara Silva, Lydia Aziato, Solina Richter, Zohra S. Lassi.

**Data curation:** Salima Meherali, Saba Nisa, Yared Asmare Aynalem, Megan Kennedy.

**Formal analysis:** Salima Meherali, Saba Nisa, Yared Asmare Aynalem.

**Investigation:** Megan Kennedy, Bukola Salami, Samuel Adjorlolo, Parveen Ali, Lydia Aziato, Zohra S. Lassi.

**Methodology:** Salima Meherali, Yared Asmare Aynalem, Megan Kennedy, Kênia Lara Silva, Zohra S. Lassi.

**Project administration:** Bukola Salami, Parveen Ali, Kênia Lara Silva.

**Software:** Megan Kennedy.

**Supervision:** Samuel Adjorlolo, Parveen Ali, Kênia Lara Silva, Lydia Aziato, Solina Richter, Zohra S. Lassi.

**Visualization:** Salima Meherali, Saba Nisa, Yared Asmare Aynalem.

**Writing – original draft:** Salima Meherali, Yared Asmare Aynalem.

**Writing – review & editing:** Saba Nisa, Bukola Salami, Samuel Adjorlolo, Parveen Ali, Kênia Lara Silva, Lydia Aziato, Solina Richter, Zohra S. Lassi.

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
