## [Decision Letter · Decision Letter 0]

25 Jun 2024

PGPH-D-24-00605

Impact of Climate Change on Maternal Health Outcomes: An Evidence Gap Map Review

Dear Dr. Meherali,

Thank you for submitting your manuscript to PLOS Global Public Health. We have received comments from two reviewers who have both suggested for revision of your manuscript. Therefore, we invite you to submit a revised version of the manuscript that addresses the points raised during the review process.

We look forward to receiving your revised manuscript.

Kind regards,

Rahul Goel, Ph.D.

Academic Editor

Journal Requirements:

4. In the online submission form, you indicated that "Data will be provided on formal request to the corresponding author". 

3. Uploaded as supplementary information.

Additional Editor Comments (if provided):

This is a well-written manuscript addressing an important global health issue. Authors should add some descriptive graphs (bar charts, for example) to highlight where evidence is available, and in doing so, also highlight where it is not. I believe that Figure 3 alone is not enough to communicate the results.

Please address the comments from the two reviewers.

Reviewers' comments:

Reviewer's Responses to Questions

**Comments to the Author**

1. Does this manuscript meet PLOS Global Public Health’s publication criteria? Is the manuscript technically sound, and do the data support the conclusions? The manuscript must describe methodologically and ethically rigorous research with conclusions that are appropriately drawn based on the data presented.

Reviewer #1: Yes

Reviewer #2: Yes

2. Has the statistical analysis been performed appropriately and rigorously?

Reviewer #1: N/A

Reviewer #2: N/A

3. Have the authors made all data underlying the findings in their manuscript fully available (please refer to the Data Availability Statement at the start of the manuscript PDF file)?

Reviewer #1: No

Reviewer #2: No

4. Is the manuscript presented in an intelligible fashion and written in standard English?

Reviewer #1: Yes

Reviewer #2: Yes

5. Review Comments to the Author

Reviewer #1: Data Availability:

The authors have stated in the Data Availability Statement that data underlying the findings in the manuscript will be provided on formal request to the corresponding author. It is recommended that the authors ensure full compliance with the PLOS Data policy by making all data fully available without restrictions, as required by the journal.

Intelligible Presentation and Language:

The manuscript is presented in an intelligible fashion and written in standard English. However, minor typographical or grammatical errors may need to be corrected during the revision process to enhance clarity and readability.

Review Comments to the Author:

The manuscript demonstrates strong methodological rigor in addressing the impact of climate change on maternal health outcomes. The authors should ensure full data availability as per PLOS guidelines. Additionally, minor language edits may be needed for clarity. No competing interests are declared.

Reviewer #2: Overall:

This is a vital area of research; it could hardly be more important. I congratulate the authors on this work, which is of high quality and highly relevant. I think the evidence mapping is a valuable contribution to the field and is worthy of publication with a relatively minor set of edits.

Introduction:

The introduction is clear and sets the context well. It might be useful to present the connections between climate change and mental health in a more structured manner, such as categorising physiological risks, social/economic risks/vulnerability, and indirect risks/harms.

Line 120: Please clarify if this refers to maternal deaths, and how conservative this estimate is, considering the potential for ambitious adaptation.

The introduction could be edited down slightly with additional structure to aid readability.

Methods:

The methods are appropriate and well-described.

It might be beneficial to illustrate the search strategy in the main text rather than just in the supplementary materials.

It would be interesting to see the proportion of grey literature in the identified papers, as I had assumed this would be significant.

Findings:

It could be beneficial to compare the trend in publications to the wider explosion in papers/publications.

Figure 5 appears to be misformatted or unclear (e.g., Africa appearing west of South America). I suggest either a histogram or a flat projection with shading to improve clarity. Currently, it slightly distracts from a core finding.

Is there any scope to look beyond the crude counts? It could be useful to disaggregate by the type of paper.

At times, the end of the findings section strays into discussion/implications.

Discussion:

Line 261: "Moreover" might not be the best word choice here as it introduces a different and important point.

Line 275: It seems to refer to a relative abundance, not an absolute one; it is worth clarifying this.

At times, the language could be clearer and more precise (e.g., Line 290).

The implications section is a little generic. It might be helpful to propose a set of specific recommendations or ideas for consideration.

Figures:

Figure 1:

The image resolution is poor.

It would be helpful to disaggregate by the type of study (e.g., grey literature vs peer-reviewed).

Figure 2:

It is challenging to effectively capture elements like the Evidence Map, but the QR code is a good idea.

It could be nice to illustrate a couple of use cases for the map, showing how the data can be utilised.

Figure 3:

The figure could be more visually engaging, possibly by illustrating the points with icons and using more colour. A redrawn version could be helpful to the reader.

Figure 5:

As previously discussed, the current projection is not appropriate. I suggest using a flat projection with coloured shading.

Supporting Information:

I have not reviewed the supporting information

---

## [Editor Report · Decision Letter 1]

10 Jul 2024

Impact of Climate Change on Maternal Health Outcomes: An Evidence Gap Map Review

PGPH-D-24-00605R1

Dear Dr Meherali,

We are pleased to inform you that your manuscript 'Impact of Climate Change on Maternal Health Outcomes: An Evidence Gap Map Review' has been provisionally accepted for publication in PLOS Global Public Health.

Best regards,

Rahul Goel, Ph.D.

Academic Editor
